# HSP_70_ Modulators for the Correction of Cognitive, Mnemonic, and Behavioral Disorders After Prenatal Hypoxia

**DOI:** 10.3390/biomedicines13040982

**Published:** 2025-04-17

**Authors:** Olena Aliyeva, Igor F. Belenichev, Ivan Bilai, Iryna Duiun, Lyudmyla Makyeyeva, Valentyn Oksenych, Oleksandr Kamyshnyi

**Affiliations:** 1Department of Histology, Cytology and Embryology, Zaporizhzhia State Medical and Pharmaceutical University, 69035 Zaporizhzhia, Ukraine; 2Department of Pharmacology and Medical Formulation with Course of Normal Physiology, Zaporizhzhia State Medical and Pharmaceutical University, 69035 Zaporizhzhia, Ukraine; 3Department of Clinical Pharmacy, Pharmacotherapy, Pharmacognosy and Pharmaceutical Chemistry, Zaporizhzhia State Medical and Pharmaceutical University, 69035 Zaporizhzhia, Ukraine; 4Faculty of Medicine, University of Bergen, 5020 Bergen, Norway; 5Department of Microbiology, Virology, and Immunology, I. Horbachevsky Ternopil National Medical University, 46001 Ternopil, Ukraine; alexkamyshnyi@gmail.com

**Keywords:** prenatal hypoxia, HSP_70_ modulators, open field test, cognitive disorders, memory disorders, behavioral disorders

## Abstract

**Background/Objectives:** Prenatal hypoxia (PH) is a leading cause of nervous system disorders in early childhood and subsequently leads to a decline in the cognitive and mnemonic functions of the central nervous system (such as memory impairment, reduced learning ability, and information processing). It also increases anxiety and the risk of brain disorders in adulthood. Compensatory–adaptive mechanisms of the mother–placenta–fetus system, which enhance the fetus’s CNS resilience, are known, including the activation of endogenous neuroprotection in response to hypoxic brain injury through the pharmacological modulation of HSP_70_. **Methods:** To evaluate the effect of HSP_70_ modulators—Cerebrocurin, Angiolin, Tamoxifen, Glutaredoxin, Thiotriazoline, and HSF-1 (heat shock factor 1 protein), as well as Mildronate and Mexidol—on the motor skills, exploratory behaviors, psycho-emotional activities, learning, and memories of offspring after PH. Experimental PH was induced by daily intraperitoneal injections of sodium nitrite solution into pregnant female rats from the 16th to the 21st day of pregnancy at a dose of 50 mg/kg. The newborns received intraperitoneal injections of Angiolin (50 mg/kg), Thiotriazoline (50 mg/kg), Mexidol (100 mg/kg), Cerebrocurin (150 µL/kg), L-arginine (200 mg/kg), Glutaredoxin (200 µg/kg), HSF-1 (50 mg/kg), or Mildronate (50 mg/kg) for 30 days. At 1 month, the rats were tested in the open field test, and at 2 months, they were trained and tested for working and spatial memory in the radial maze. **Results:** Modeling PH led to persistent impairments in exploratory activity, psycho-emotional behavior, and a decrease in the cognitive–mnestic functions of the CNS. It was found that Angiolin and Cerebrocurin had the most pronounced effects on the indicators of exploratory activity and psycho-emotional status in 1-month-old animals after PH. They also exhibited the most significant cognitive-enhancing and memory-supporting effects during the training and evaluation of skill retention in the maze in 2-month-old offspring after PH. **Conclusions:** for the first time, we obtained experimental data on the effects of HSP_70_ modulators on exploratory activity, psycho-emotional behavior, and cognitive–mnestic functions of the central nervous system in offspring following intrauterine hypoxia. Based on the results of this study, we identified the pharmacological agents Angiolin and Cerebrocurin as promising neuroprotective agents after perinatal hypoxia.

## 1. Introduction

The course of intrauterine development and the neonatal period largely determine the future health status and quality of life of an individual. Prenatal hypoxia (PH) is the main cause of prenatal cerebral pathology and neonatal mortality [1,2,3]. In the structure of neonatal mortality, hypoxia ranks second after prematurity. Approximately 12–23% of infant mortality worldwide (0.7–1.2 million annually) is registered as a result of prenatal hypoxia, with 0.5 million children becoming disabled [4,5]. Doctors have specific medications for treating the consequences of hypoxic-ischemic damage to the central nervous system. However, the high mortality and disability rates in children indicate that these pharmacological agents are not fully effective. Therefore, a priority task in modern pharmacology is the development of new approaches for the pharmacological correction of neurological disorders resulting from PH [6,7,8]. In most cases, the causes of the fetus’s hypoxic condition are placental insufficiency, maternal disease, and infectious–inflammatory processes [9]. The leading mechanisms of disorders resulting from prenatal hypoxia are excitotoxicity, oxidative stress, and inflammation [10,11]. Glutamate excitotoxicity leads to nitrosative stress [12,13]. The toxic byproducts of nitric oxide metabolism, formed in cells with the involvement of nNOS and iNOS, along with reactive oxygen species (ROS), bind to important macromolecules, damaging cells and activating the signaling molecules of apoptosis [14,15]. Such disturbances resulting from intrauterine hypoxia lead to neurological disorders and impairments in the nervous system in early childhood, reduced learning and memory abilities, anxiety, and an increased risk of developing mental and neurodegenerative diseases in later stages of life [16,17,18].

Neurons have the ability to adapt to changes caused by the chemical and physical factors acting on them. The adaptation of neurons to hypoxia at the cellular and subcellular levels is regulated by the specific transcription factor HIF-1 [19,20,21]. The ultimate result of HIF-1 activation is an increase in oxygen delivery to the cells [22,23]. One of the primary genomic responses of cells to hypoxia and energy stress is the induction of heat shock protein (HSP), whose effect on stabilizing HIF-1 ensures the activation of processes such as proliferation, apoptosis, and angiogenesis under ischemic conditions [24,25]. Various studies, including ours, have demonstrated the compensatory–adaptive mechanisms of the mother–placenta–fetus system that enhance the fetal central nervous system’s resilience. Among these mechanisms is the activation of endogenous neuroprotection in response to hypoxic brain injury, with HSP_70_ playing an active role. Additionally, pharmacological modulation of HSP_70_ is shown to be a promising direction for neuroprotection following prenatal hypoxia [26,27].

Our data demonstrate the effectiveness of HSP_70_ modulators—Cerebrocurin, Angiolin, Tamoxifen, Glutaredoxin, Thiotriazoline, and HSF-1 (heat shock factor 1 protein), as well as Mildronate and Mexidol—in experimental neuroprotection following prenatal hypoxia. The use of these agents led to a reduction in mortality during the early postnatal period, a decrease in the markers of oxidative and nitrative stress, and an increase in the expression and concentration of HSP_70_ and HIF-1 in the cytosolic and mitochondrial fractions of the brain in animals after PH [28,29,30]. There is evidence of the protective effect of HSP_70_ on higher CNS functions under various adverse influences on the brain [31,32]. All of this has served as the experimental basis for further preclinical studies of HSP_70_ modulators in prenatal hypoxia, specifically to investigate their impact on cognitive, memory, and behavioral disorders in offspring.

Objective of this study: to assess the impact of HSP_70_ modulators—Cerebrocurin, Angiolin, Tamoxifen, Glutaredoxin, Thiotriazoline, and HSF-1 (heat shock factor 1 protein), as well as Mildronate and Mexidol—on motor activity, exploratory behavior, psycho-emotional activity, learning, and memory performance in offspring following PH.

## 2. Materials and Methods

### 2.1. Experimental Model

This study was conducted in accordance with Directive 2010/63/EU of the European Parliament and Council of 22 September 2010 on the protection of animals used for scientific purposes, as well as the national “General Ethical Principles of Animal Experimentation” (Ukraine, 2001) and the guidelines outlined in “Basic Methods for Studying the Toxicity of Potential Pharmacological Drugs” (HEC of Ukraine, Kyiv, 2000). This experiment was approved by the Bioethics Commission of Zaporizhzhia State Medical University (protocol No. 32 dated 16 November 2021).

This experimental study was carried out on 50 female and 10 male white rats weighing 220–240 g and aged 4.5 months, obtained from the vivarium of the State Enterprise Institute of Pharmacology and Toxicology of the National Academy of Medical Sciences of Ukraine. Before this experiment, each animal was examined by a qualified veterinarian to assess its health status. The animals then underwent a 10-day quarantine period and were randomly assigned to groups based on the dose of the administered drug. The animals were placed in polycarbonate cages measuring 550 × 320 × 180 mm, with galvanized steel lids measuring 660 × 370 × 140 mm, that contained glass drinking bottles. No more than five rats were housed in each cage. Each cage was labeled with a tag indicating the study number, species, sex, animal numbers, and dosage. The cages were arranged on shelves according to dose levels and cage numbers specified on the labels.

The animal housing room maintained the following conditions: a temperature of 20–24 °C, a humidity level of 30–70%, and a lighting cycle of 12 h light/12 h dark. All rats were fed ad libitum a standard diet for laboratory animals provided by the company “Phoenix” (Kharkiv, Ukraine). Water from the municipal water supply (subjected to reverse osmosis and sterilized by UV radiation) was provided without restriction. Wood shavings from alder (*Alnus glutinosa*), autoclaved prior to use, were used as bedding. A model of chronic hypoxia induced by sodium nitrite during pregnancy was used for the experiment, which was reproduced by daily intraperitoneal administration of sodium nitrite solution to pregnant female rats from days 16 to 21 of pregnancy at a dose of 50 mg/kg [29]. Modeling PH by administering sodium nitrite to pregnant females leads to hemic hypoxia due to the formation of methemoglobin, which combines with tissue hypoxia due to the uncoupling of oxidation and phosphorylation processes. Disruption of oxygen transport function in the blood of pregnant female rats results in impaired uteroplacental circulation and oxygen starvation of the fetus or embryo. This, in turn, leads to hypoxic encephalopathy of varying severity in adult individuals when exposed to hypoxic factors during the prenatal period of development [33]. In the control group, pregnant females were administered physiological saline in the same regimen.

The offspring were divided into groups as follows:Group 1: healthy animals from females with a physiologically normal weight, administered physiological saline intraperitoneally at a volume of 5 µL/g;Group 2: control group after prenatal hypoxia (PH), administered physiological saline intraperitoneally at a volume of 5 µL/g;Group 3: after PH, administered Angiolin intraperitoneally at a dose of 50 mg/kg;Group 4: after PH, administered Piracetam intraperitoneally at a dose of 500 mg/kg;Group 5: after PH, administered Tamoxifen intranasally at a dose of 0.1 mg/kg;Group 6: after PH, administered Thiotriazoline intraperitoneally at a dose of 50 mg/kg;Group 7: after PH, administered Mexidol (nicomex) intraperitoneally at a dose of 100 mg/kg;Group 8: after PH, administered Cerebrocurin intraperitoneally at a volume of 150 µL/kg (or at a dose of 0.3 mg/kg, calculated based on active neuropeptides);Group 9: after PH, administered L-arginine intraperitoneally at a dose of 200 mg/kg;Group 10: after PH, administered Glutaredoxin intraperitoneally at a dose of 200 µg/kg;Group 11: after PH, administered HSF-1 intraperitoneally at a dose of 50 mg/kg;Group 12: after PH, administered Mildronate intraperitoneally at a dose of 50 mg/kg.

The following pharmaceutical preparations and pharmacological agents were used in this study:Thiotriazoline (morpholinium 3-methyl-1,2,4-triazolyl-5-thioacetic acid, 2.5% solution for injections, Arterium Corporation, Kyiv, Ukraine);Tamoxifen (tablets, Finland, on the basis of which an extemporaneously prepared intranasal gel (1 mg/1 mL) was made at the Department of Pharmaceutical Technology, ZSMU);Angiolin ([(S)-2,6-diaminohexanoic acid 3-methyl-1,2,4-triazolyl-5-thioacetate], NPO Pharmatron, Odesa, Ukraine);Glutaredoxin (Sigma-Aldrich, St. Lous, MO, USA);Cerebrocurin (injection solution containing neuropeptides, S-100 proteins, rylin, nerve growth factor (NGF) (no less than 2 mg/mL), and amino acids, NPO NIR, Kharkiv, Ukraine);L-Arginine (42% solution for injections in vials, Tivortin, Yuriy-Pharm, Kyiv, Ukraine);Mexidol (2-ethyl-6-methyl-3-hydroxy-pyridyl succinate, injection solution, 50 mg/mL, NPK Pharmasoft, Ellara LLC, Moscow, Russia);Mildronate (2-(2-carboxyethyl)-1,1,1-trimethylhydrazinium, 10% solution for injections in ampoules, Grindex, Riga, Latvia)Piracetam (200 mg/mL, JSC Pharmak, Kyiv, Ukraine);HSF-1 (Sigma-Aldrich, St. Lous, MO, USA).

All the offspring used were identical in weight and gestation period. The choice of doses was justified either by our experimental studies or by other authors [34,35,36,37,38].

Behavioral and exploratory activity of the animals was assessed one month after birth, immediately after the experimental therapy course, while the evaluation of cognitive and mnemonic functions of the central nervous system was conducted at 2 months of age (one month after the experimental therapy course).

### 2.2. Assessment of Motor and Exploratory Activity

Assessment of motor and exploratory activity was conducted using the “open field” method with an arena of our own design measuring 80 × 80 × 35 cm [38]. The animal was placed in the center of one of the sides with its nose facing the wall, after which it was allowed to move freely within the arena for 8 min. We assessed the total distance traveled (cm), overall motor activity (cm^2^/s), activity structure (high activity, low activity, inactivity, %), number of freezing episodes, entries into the center, distance traveled along the wall (cm) and in the central area of the arena (cm, %), vertical exploratory activity (number of rears on the hind legs at the wall and in the center), number of short and long grooming events, and the number of defecation and urination acts.

### 2.3. Assessment of Reference and Working Memory

The rats were subjected to food deprivation. Food was available daily for 1 h. The animals were reduced to 85% of their original body weight by restricting their food intake while allowing free access to water.

Memory was assessed using a radial maze (LE760, AgnTho’s, Lund, Sweden) [39]. The eight-arm radial maze consists of an octagonal platform (side length of 22 cm), with eight numbered radial arms (1 to 8) extending from the platform. Each arm is 70 cm long and 10 cm wide, with a food cup at the end (diameter 2 cm, depth 1.5 cm). Each arm can be closed independently using a guillotine mechanism. The entire setup was positioned 70 cm above the floor. We used the methodology as described previously [40]. The experiment was conducted in complete silence.

Starting on the first day, the animals were placed on the central platform of the maze, with 4 closed arms and 4 open arms, each containing 200 mg of food pellets in the food cups. The combination of open and closed arms was unique and constant for each animal. Over the next 10 days, the animals were trained to locate the food using external visual cues. The training session lasted for 10 min or until the animal found all four food sources. The experiment was repeated twice daily with each animal. After the experiment, the animal was given its daily food ration.

On the 10th day, the animal was placed in the radial maze with all eight arms open, with food placed in four arms according to the usual pattern for that animal. We assessed reference memory (the long-term knowledge of the maze structure and food locations that the animal developed during the training process) and the number of reference memory errors (the first visit to a previously closed arm where the animal had never found food). We also assessed working memory (the short-term representation of the food locations in the specific trial) and the number of working memory errors (revisiting an arm where the animal had previously found or not found food). Additionally, we evaluated the total distance traveled and overall motor activity.

### 2.4. Data Collection and Processing

This research was conducted at the Department of Experimental Pathophysiology and Functional Morphology of the Educational Medical Laboratory Center of Zaporizhzhia State Medical University. The experiments were performed in a well-lit room in complete silence. External and internal visual, olfactory, and auditory stimuli were excluded during the experiments. Animal behavior was assessed by a technician who was unaware of the animal’s group assignment. Image capture and recording were performed using a color video camera (SSC-DC378P, Sony, Tokyo, Japan). Video file analysis was conducted using Smartv 3.0 software (Harvard Apparatus, Holliston, MA, USA). Statistical processing of the results was performed using Microsoft Excel 2016 with the AtteStat 12 statistical analysis package. The Kruskal–Wallis test with Dunn’s post hoc correction was used to assess the significance of differences between experimental groups. Differences were considered statistically significant at *p* < 0.05.

## 3. Results

When assessing the specific indicators of the open field test, it was found that modeling intrauterine hypoxia negatively affected the behavioral characteristics of the animals (Table 1, Table 2 and Table 3). Intrauterine hypoxia led to a 1.33-fold decrease (*p* < 0.05) in the overall activity of 1-month-old animals compared to 1-month-old animals born after physiological pregnancy. Additionally, hypoxia resulted in a twofold reduction (*p* < 0.05) in the distance traveled, alongside a 2.7-fold increase (*p* < 0.05) in immobility and a 1.85-fold increase (*p* < 0.05) in immobility compared to the intact group (healthy 1-month-old rats) (Table 1 and Table 2).

Furthermore, in the animals that experienced intrauterine hypoxia, there was a 1.86-fold increase (*p* < 0.05) in the number of freezing behaviors, a 2.2-fold increase (*p* < 0.05) in the time spent in the illuminated center, and a 1.46-fold increase (*p* < 0.05) in the distance traveled in the darkened area (near the wall) (Table 1). These findings indicate reduced motor activity in the post-hypoxia animals and the development of anxiety, fear, and excitability.

Modeling intrauterine hypoxia led to a twofold increase (*p* < 0.05) in wall-posting in the 1-month-old animals compared to the healthy rats of the same age. Additionally, in the control group animals, there was a fourfold decrease (*p* < 0.05) in the number of short grooming acts, accompanied by a threefold reduction (*p* < 0.05) in defecation acts (Table 2). This also points to increased anxiety, excitability, irritability, and a reduced sense of comfort in the animals.

In the control group, a 4.14-fold decrease (*p* < 0.05) in high activity was observed, suggesting the high emotionality and excitability of the animals. The reduction in high activity may also be interpreted as a decreased ability to engage in exploratory and searching behavior, as the rat becomes inactive, is fearful of entering the illuminated space, and spends more time adapting to the new environment.

When assessing the specific indicators of learning and memory reproduction in the radial arm maze in 2-month-old rats after intrauterine hypoxia, it was found that 10 days after the learning process, the animals that had experienced hypoxia exhibited persistent cognitive and mnemonic impairments compared to the 2-month-old animals born after physiological pregnancy. During the reproduction of the learning results in the 2-month-old rats after hypoxia, it was established that 10 days after the last training session, there was a 3.2-fold increase in the number of working memory errors and a twofold increase in the number of reference memory errors (Table 3). These findings indicate a disruption of higher CNS functions after intrauterine hypoxia.

The introduction of HSP_70_ modulators, antioxidants, nootropics, and metabolotropic therapy agents immediately after birth to the experimental animals following PH had varying effects on the main indicators of higher CNS functions, with some showing no improvement in behavioral reactions, emotional status, or cognitive–mnestic functions. Analyzing the results of the experimental studies, it was found that Angiolin and Cerebrocurin had the most pronounced effects on the indicators of exploratory activity and psycho-emotional status in 1-month-old animals after PH, as observed in the open field test. Other drugs and pharmacological agents, such as Piracetam, Mildronate, Glutaredoxin, Thiotriazoline, and HSF-1, had a moderate effect on the specific indicators of exploratory activity and psycho-emotional status in 1-month-old animals after PH. The drugs Mexidol, Tamoxifen, and L-Arginine showed no effect on the studied indicators in the open field test.

The cumulative administration of Cerebrocurin immediately after the discontinuation of the treatment led to a 2.5-fold increase (*p* < 0.05) in the overall motor activity of the animals compared to the control group (untreated animals) and a 1.87-fold increase (*p* < 0.05) compared to the group of healthy animals (born after physiological pregnancy). In the experimental animals treated with Cerebrocurin, the free locomotion distance increased 2.6 times (*p* < 0.05) compared to the control group and 1.3 times (*p* < 0.05) compared to the healthy animal group, while high activity was increased 4.2 times (*p* < 0.05) compared to the control group. This indicated a high level of motor and exploratory activity in animals after PH who were treated with Cerebrocurin. These animals were active, mobile, and intensively explored new spaces.

Cerebrocurin also positively affected the psycho-emotional status of the 1-month-old animals after PH. The animals actively entered the illuminated arena, did not linger in the dark and semi-dark areas, actively groomed, and displayed reduced anxiety and aggression. In this experimental group, the number of freezing behaviors decreased 1.8 times (*p* < 0.05), inactivity decreased 1.57 times (*p* < 0.05), the time spent in the illuminated center of the arena decreased 1.86 times (*p* < 0.05), the distance traveled in the darkened area decreased 1.3 times (*p* < 0.05), and the total immobility score decreased 2.47 times (*p* < 0.05) compared to the control group. The introduction of Cerebrocurin to animals after PH resulted in the complete restoration of psycho-emotional behavior indicators (such as grooming, defecation acts, and wall-posting behavior) to the level of the intact group (healthy animals).

The 1-month-old animals that underwent PH and were treated with Angiolin immediately after birth also showed no difference in exploratory activity compared to the healthy animals of the same age. These animals were active, with overall motor activity increasing 2.2 times (*p* < 0.05) compared to the control group and 1.63 times (*p* < 0.05) compared to the healthy animal group. The experimental animals treated with Angiolin traveled 2.5 times more free distance (*p* < 0.05) compared to the control group and 1.24 times (*p* < 0.05) compared to the healthy animals, and high activity increased 3.4 times (*p* < 0.05) compared to the control group.

Angiolin, similar to Cerebrocurin, positively influenced the psycho-emotional behavior of the 1-month-old animals after PH. Not only did it reduce aggression and anxiety, but it also decreased the fear of entering a new illuminated space and increased comfort and empathy. In the group of animals treated with Angiolin, freezing behaviors decreased 2 times (*p* < 0.05), inactivity decreased 2.2 times (*p* < 0.05), the time spent in the illuminated center decreased 2.3 times (*p* < 0.05), the distance traveled in the darkened area decreased 1.4 times (*p* < 0.05), and overall immobility decreased 1.86 times (*p* < 0.05) compared to the control group. The course of administration of Angiolin to animals after PH proved to be as successful as Cerebrocurin, leading to the restoration of psycho-emotional behavior indicators (grooming, defecation acts, and wall-posting behavior) to the intact level (healthy animals).

The pharmaceutical drugs and pharmacological agents—Piracetam, Mildronate, Glutaredoxin, Thiotriazoline, and HSF-1—significantly increased overall activity, the distance traveled, and significantly reduced the immobility score (HSF-1, Piracetam, Mildronate (*p* < 0.05)) in animals after perinatal hypoxia (PH). This indicates the restoration of motor activity in the experimental animals during the exploration of a new environment. However, these pharmacological agents had no significant effect on psycho-emotional behavior (anxiety, aggression, fear, discomfort) or the exploratory component of behavior.

It is also worth noting that the intranasal administration of Tamoxifen influenced fear and anxiety by reducing the number of freezing (*p* < 0.05) and wall-posting (*p* < 0.05) behaviors.

The evaluation of the effects of the studied pharmacological agents on the cognitive–mnestic functions of the CNS in 2-month-old animals after PH revealed the following. The most pronounced mnemonic and nootropic effects were observed with Angiolin and Cerebrocurin. Thirty days after the completion of the course of administration of these agents, the number of reference memory errors in the animals that underwent PH decreased by four times (*p* < 0.05) and two times (*p* < 0.05), respectively. Also, after the course of administration of Angiolin and Cerebrocurin, the number of working memory errors decreased by 2.28 times (*p* < 0.05). The other experimental agents—Mexidol, Mildronate, Thiotriazoline, Glutaredoxin, Tamoxifen, Piracetam, L-Arginine, and HSF-1—did not significantly affect the number of reference memory errors made by the experimental animals in the radial arm maze after training. However, Piracetam, Glutaredoxin, Thiotriazoline, and HSF-1 significantly reduced the number of working memory errors in the radial arm maze after training in the animals that underwent PH.

Thus, it was established that modeling PH led to persistent impairments in exploratory activity, psycho-emotional behavior, and cognitive–mnestic functions of the CNS in animals. The HSP_70_ modulators, especially Angiolin and Cerebrocurin, had a positive effect on overall motor activity, exploratory behavior, reducing anxiety, irritability, aggression, and fear, and contributed to the improvement of learning and the retention of acquired skills.

## 4. Discussion

The data we obtained align with the concept of CNS impairments resulting from prenatal hypoxia (PH). PH leads to an increased risk of developing psychomotor and cognitive–mnestic disorders, which are often not compensated for throughout the subsequent life [40]. We have demonstrated an increase in offspring mortality following PH [29]. A number of researchers have noted that following PH, animals often exhibit motor and learning impairments and hyperactivity syndrome, as well as reduced memory, socialization, and attention [41]. The role of PH in the development of brain functions during the postnatal period and the subsequent increased risk of neurodegenerative disorders later in life has been described. PH has a significant impact on the expression and processing of various genes involved in brain development. It leads to alterations in the expression patterns of mRNA and proteins such as acetylcholinesterase, amyloid precursor protein, and amyloid-β peptide. The disruption of their expression and metabolism caused by PH may contribute not only to early cognitive dysfunctions but also to the development of neurodegeneration in later life [42]. One of the main causes of cognitive disfunctions after PH is considered to be disturbances in energy metabolism, including the lack of energy in the synapse to maintain glutamatergic transmission, neurotransmitter reuptake, the inhibition of synaptogenesis, and the increased production of ROS through neurochemical and energetic reactions, as well as the initiation of mechanisms of circulatory ischemia (hypoperfusion, changes in brain perfusion, intracranial pressure) [43,44,45,46]. PH leads to impairments in various types of memory (short-term memory, long-term memory, working memory, and reference memory) in the offspring of rats. This is associated with structural changes observed in the hippocampus due to the activation of apoptosis and neuronal cell death [47,48,49]. PH leads to the development of persistent cognitive deficits, as well as psycho-emotional disturbances such as sluggishness, fear, anxiety, disorientation, aggression, and irritability. Several studies have shown that PH results in behavior similar to anxiety and depression, such as reduced social interaction, impaired sexual behavior in males, and a decrease in the acquisition of both passive and active avoidance responses [44,50]. Thus, in 1-month-old rats after PH, basal synaptic transmission in the CA3-CA1 synapses was significantly impaired, and long-term synaptic potentiation in the hippocampus was reduced twofold. Additionally, there was a decrease in the affinity of the GluN2B receptor, accompanied by a significant deficit in learning and memory [51]. It has been established that PH led to a significant increase in the production of the cytotoxic products of nitrosative stress due to the elevated expression of iNOS and nNOS, in connection with the deprivation of the thiol–disulfide system intermediates, in the CA1 hippocampi of 1- and 2-month-old rats that underwent intrauterine hypoxia [14,42].

It was also found in this age group of experimental animals that PH led to a decrease in the expression of HIF-1α, HSP_70_, and c-fos mRNA, as well as the concentration of HSP_70_ and HIF-1α in the brain and plasma, and an increase in offspring mortality. We established a direct correlation between the levels of HSP_70_ and the survival of the offspring [28,52]. HSP_70_ is important for memory formation, as it facilitates the folding and transport of synaptic proteins, modulates the signaling cascades associated with synaptic activation, and is involved in the mechanisms of neurotransmitter release [53]. c-fos plays a role in the development of cognitive and mnemonic dysfunctions of the CNS after TBI. c-fos is actively involved in the mechanisms of endogenous neuroprotection, angiogenesis, memory consolidation, and the expression of heat shock protein genes (HSP_70_ and HSP_72_) as well as hypoxia-inducible factor (HIF-1). TBI alters the expression of HIF-1α in the brains of embryonic rats depending on the stage of embryonic development, followed by a sustained increase and activation of the HPA axis, causing hypoxic disease and behavioral changes [50,54,55,56]. It is known that the inhibition of c-fos mRNA translation in brain structures impairs short-term memory in various learning models across different animal species [57,58,59,60,61] (Figure 1).

Presumably, the leading mechanisms of impaired CNS cognitive–mental functions in offspring after PH are disturbances in energy metabolism, ATP deficiency leading to transmitter autocoidosis, reduced energy supply for synaptic transmission, and the formation of secondary mitochondrial dysfunction. The next step is an increased production of ROS and NO by energetic and neurochemical reactions and the triggering of the mechanisms of circulatory ischemia. Mitochondrial dysfunction and nitrosative and oxidative stresses initiate the trigger mechanisms of neuroinflammation and neuronal apoptosis, leading to impaired mechanisms of short- and long-term memory. Excess ROS on the background of a deprivation of the antioxidant system and the factors of endogenous neuroprotection (HSP_70_ and HIF-1) leads to the desensitization of receptors (GABA, serotonin, glutamine), suppression of transporter proteins expression, and an increase in psycho-emotional disorders—lethargy, fear, anxiety, disorientation, aggressiveness, and irritability.

In previous studies, we showed that the greatest impact on the expression of HSP_70_ and HIF-1α in the brains of animals after PH was exerted by Cerebrocurin and Angiolin. Agents such as HSF-1, Thiotriazoline, Glutaredoxin, and Tamoxifen moderately affected HSP_70_ levels, while L-Arginine, Mexidol, and Piracetam showed minimal or no effect. These studies also demonstrated that the course of administration of the investigated pharmacological agents starting from the first day of life reduced offspring mortality following prenatal hypoxia. The highest survival rates were observed in the groups of rats treated with Mildronate, Angiolin, Thiotriazoline, HSF-1, and Cerebrocurin [29,34]. Thus, there is a certain correlation between the effect on the levels of HSP_70_ and HIF-1α and the cognitive-enhancing action of the drugs. The properties of Cerebrocurin we identified are consistent with the findings of other researchers, who demonstrated the high antioxidant activity of the drug. They determined that Cerebrocurin increased the expression of Mn-SOD, and they studied its neuroprotective and anti-ischemic activity, realized through direct mitoprotection, the increased synthesis of HSP and HIF proteins, and their mediated activation of compensatory energy shunts under acute cerebral ischemia conditions [52]. Cerebrocurin contains peptides that carry the program for analyzing the condition and construction of the CNS. Therefore, the final effect of Cerebrocurin varies due to its qualitatively distinct mechanism of action compared to other neuroprotectors. The protective effects of Cerebrocurin on brain tissue include its optimizing action on brain energy metabolism and calcium homeostasis and its stimulation of the intracellular synthesis of protective proteins, growth factors for neurons, and neurotrophic factors, as well as the slowing down of glutamate-calcium cascade processes and oxidative protein modification. Cerebrocurin may reduce disturbances in higher CNS functions due to alcohol intoxication, hypoxia, and cerebral circulation disorders. This action of Cerebrocurin is linked to its optimizing effect on the expression of HSP_70_ and c-fos. The reduction in anxiety and aggressiveness in the offspring treated with Cerebrocurin is likely associated with its ability to influence the mechanisms of neurotransmitter reuptake and reduction of the desensitization of the corresponding receptors [60,61] (Figure 2).

Cerebrocurin contains peptides that carry a program to analyze the state and structure of the CNS. Cerebrocurin, through its effect on the expression of HSP 70, triggers regulatory mechanisms aimed at increasing the expression of antioxidant enzymes, the expression of HIF proteins, the activation of compensatory mitochondrial–cytosolic shunts of energy production and calcium homeostasis, the stimulation of the intracellular synthesis of protective proteins, neuronal growth factors, and neurotrophic factors, the increase in the affinity of BDNF binding to its receptors, and the normalization of c-fos expression. Cerebrocurin exhibits a significant antiapoptotic effect. Cerebrocurin positively affects the mechanisms of transmitter reuptake through the expression of transporter proteins.

Significant improvements in exploratory, motor, cognitive-enhancing, and mnemonic activities in the offspring after TBI, as well as the restoration of their psycho-emotional behavior following the administration of Angiolin, are ensured by the following mechanisms. First, through its antioxidant mechanism (scavenging of ROS and NO), Angiolin exerts a positive effect on the expression of genes encoding the synthesis of superoxide dismutase, glutathione peroxidase, and glutathione reductase. It regulates ROS/GSH-dependent mechanisms of HSP_70_ expression and, through this, promotes the prolongation of the HIF-1 lifespan. Angiolin also regulates the transport of synaptic proteins, modulates signaling cascades related to the reverberation of excitation in neuronal circuits, and reduces the oxidative modification of protein molecules, including receptors and ion channels [34,53]. Secondly, Angiolin, under conditions of brain ischemia, reduces the number of damaged mitochondria in the cortex and CA1 hippocampus. It is capable of activating compensatory cytosolic–mitochondrial energy shunts and increasing ATP production, which are necessary to sustain memory protein synthesis and synaptic function [34]. Many authors confirm the importance of improving brain energy metabolism in the fight against dementia [62,63]. It is also known that mitochondrial dysfunction is a hallmark of many diseases that impair brain function and affect cognitive abilities [61], while the improvement of cognitive and mnemonic functions is associated with mitoprotection [64]. All of this allows us to evaluate the cognitive-enhancing action of Angiolin through the prism of its mitoprotection. Thirdly, Angiolin, due to the L-lysine residue in its composition, increases the affinity of GABA receptors and reduces excitotoxicity [65,66]. The increase in GABA receptor affinity under the action of Angiolin may explain its anxiolytic effect (reduction of anxiety, fear, and aggressiveness) (Figure 3).

The pharmacological properties of Angiolin are determined by the conversion of L-lysine into pipecolic acid, which increases the affinity of the GABA–benzodiazepine receptor complex and thus reduces manifestations of glutamate excitotoxicity. Angiolin preserves the ultrastructural and functional activity of mitochondria, increases ATP production, and activates compensatory energy shunts. Angiolin activates the glutathione link of the thiol–disulfide system and thereby regulates the ROS/GSH-dependent mechanisms of HSP_70_ expression, which contributes to the increase in the HIF-1 lifespan, regulates the transport of synaptic proteins, modulates the signaling cascades associated with the reverberation of excitation in neuronal chains, and reduces the oxidative modification of protein molecules of receptors and ion channels. Angiolin exhibits NO scavenger properties, inhibits nitrosative stress reactions, and has anti-apoptotic properties.

The activation of HSP_70_ transcription in all the organisms that have been studied primarily occurs through the activation of members of the HSF family [64,67]. The results we obtained regarding the positive influence of HSF1 on the activity of offspring after TBI, particularly in novel space exploration and learning, are consistent with other studies. It is known that HSF1 is involved in memory formation processes. The activation of HSF1 leads to improved cognitive abilities, while its deficiency, along with the deficiency of HSP_70_, is associated with neurodegeneration [55,68]. In addition to regulating the HSP system, HSF-1 is involved in the expression of many genes and regulates processes such as metabolism, RNA splicing, apoptosis, ubiquitination, protein degradation, and intracellular signaling [69,70,71]. HSF-1 leads to the inhibition of TNFα-induced apoptosis, including in neurons, thereby preserving higher CNS functions [72,73,74,75,76,77,78,79]. In this study, Glutaredoxin demonstrated a positive effect on the motor and cognitive activities of offspring after PH and positively influenced learning in the maze. We explain this effect from the perspective of its impact on the GSH-dependent pathways of endogenous neuroprotection and the increased expression of mRNA for HSP_70_, HIF-1α, and HIF-3α under conditions of acute experimental cerebral ischemia, as well as the enhancement of glutathione synthesis [80,81,82,83,84,85,86,87,88,89,90]. It is known that the role of the glutathione component of the thiol–disulfide system in the brain is crucial for maintaining the cognitive functions of the CNS [91].

Thiotriazoline, in this study, demonstrated satisfactory activity in restoring motor and exploratory activities in a novel environment in the offspring after prenatal glucocorticoid exposure, as well as a positive effect on their learning ability and skill acquisition. Such effects of Thiotriazoline are not new and have been described in various studies. The ability of Thiotriazoline to modulate the antioxidant, nootropic, mnemonic, and antiamnesic activities of Piracetam has also been demonstrated, which served as the basis for the development of the drug Tiocetam (Thiotriazoline + Piracetam) [35,92]. The mnemonic and cognitive-enhancing effects of Thiotriazoline can be explained in terms of its antioxidant mechanism. Due to the specific structure of its molecule, Thiotriazoline acts as a scavenger of NO and ROS, thereby regulating the expression of NF-kB, which may provide a significant antioxidant effect as well as regulation of the expression of protective proteins. The mechanism by which Thiotriazoline affects the expression of HSP_70_ has been well described in several studies and is associated with the SH/SS regulation of transcription factor activity. Through its positive influence on mitochondria, Thiotriazoline improves energy metabolism and enhances the performance, motor activity, and synaptic transmission of excitation [24,36,52,61]. The observed effect of Thiotriazoline on cognitive, cognitive-enhancing, and mnemonic activity in the offspring after prenatal glucocorticoid exposure is attributed to its properties as a scavenger of ROS and a regulator of their concentration. The physiological concentration of ROS, particularly superoxide, modulates synaptic plasticity at the cellular level and plays a crucial role in the formation and maintenance of memory [93]. A significant increase in ROS levels leads to the disruption of the neural networks essential for memory function [33,37,94,95,96].

In our study, the use of Tamoxifen as a modulator of HSP_70_ to improve behavioral, exploratory, and cognitive activities in the offspring after prenatal glucocorticoid exposure did not meet expectations. A positive effect of Tamoxifen was observed in the reduction of anxiety in the 1-month-old offspring. It appears that the influence on HSP_70_-dependent mechanisms of memory formation and preservation by Tamoxifen was insufficient [97,98,99,100,101,102,103,104]. The anxiolytic effect of Tamoxifen is likely related to its modulatory action on central estrogen receptors. Activation of ERβ may contribute to the generation of anxiolytic effects [89]. The anxiolytic effect of Tamoxifen in our study can also be explained by the activation of the GABAergic system through the modulation of ERβ [105,106]. It is also possible that the enhancement of Tamoxifen’s anxiolytic effect is linked to its ability to influence glutathione levels [107,108,109,110,111,112]. Thus, for the first time, we obtained experimental data on the effects of HSP_70_ modulators on exploratory activity, psycho-emotional behavior, and cognitive–mnestic functions of the central nervous system in offspring after intrauterine hypoxia. As a result of this study, we identified the pharmacological agents Angiolin and Cerebrocurin as promising neuroprotective agents for treatment after PH.

## 5. Perspectives for Further Research

Further studies are planned to investigate the effects of HSP_70_ modulators on the morpho-functional characteristics of neurons in the sensorimotor cortex and CA1 hippocampus of offspring after perinatal hypoxia. Additionally, comprehensive preclinical trials of the developed nasal gel of Angiolin are planned.

## 6. Limitations of This Study

Several key limitations should be acknowledged.

The pharmacological agents were administered at a single dose, which limits the understanding of the dose–response relationships. This highlights the need for further studies to determine the optimal dosing regimen for this age group and to identify the potential side effects associated with the prolonged administration of higher doses.

This study focused on the effects of pharmacological agents administered during the first month of life and evaluated the outcomes only within that same time frame. As a result, the long-term persistence of the protective effects remains uninvestigated.

This study did not account for potential sex-based differences in the response to the pharmacological treatments, which limits the generalizability of the findings. Additionally, the use of only rat models constrains the extrapolation of the results to human conditions without further validation in other species.

The assessment of the protective effects on higher CNS functions was limited to the open field test and radial maze learning. This underscores the need for complementary research involving morphometric, immunohistochemical, molecular, and physiological analyses to more thoroughly substantiate the improvements in cognitive and mnemonic functions, as well as psycho-emotional behavior, in offspring following PH.

## 7. Conclusions

Modeling PH leads to persistent impairments in the exploratory activity and psycho-emotional behavior of animals, as well as a decrease in the cognitive–mnestic functions of the CNS.The administration of HSP_70_ modulators, antioxidants, nootropics, and metabolotropic therapy agents immediately after birth to experimental animals following PH had varying effects on the main indicators of higher CNS functions, either failing to improve behavioral responses, emotional status, or cognitive–mnestic functions, or showing effects in a specific direction and with varying intensity.The results obtained provide an experimental justification for the clinical use of Cerebrocurin and, especially, Angiolin for the correction of anxious behavior, excitability, and cognitive–mnestic dysfunction in offspring after PH.

## Figures and Tables

**Figure 1 biomedicines-13-00982-f001:**
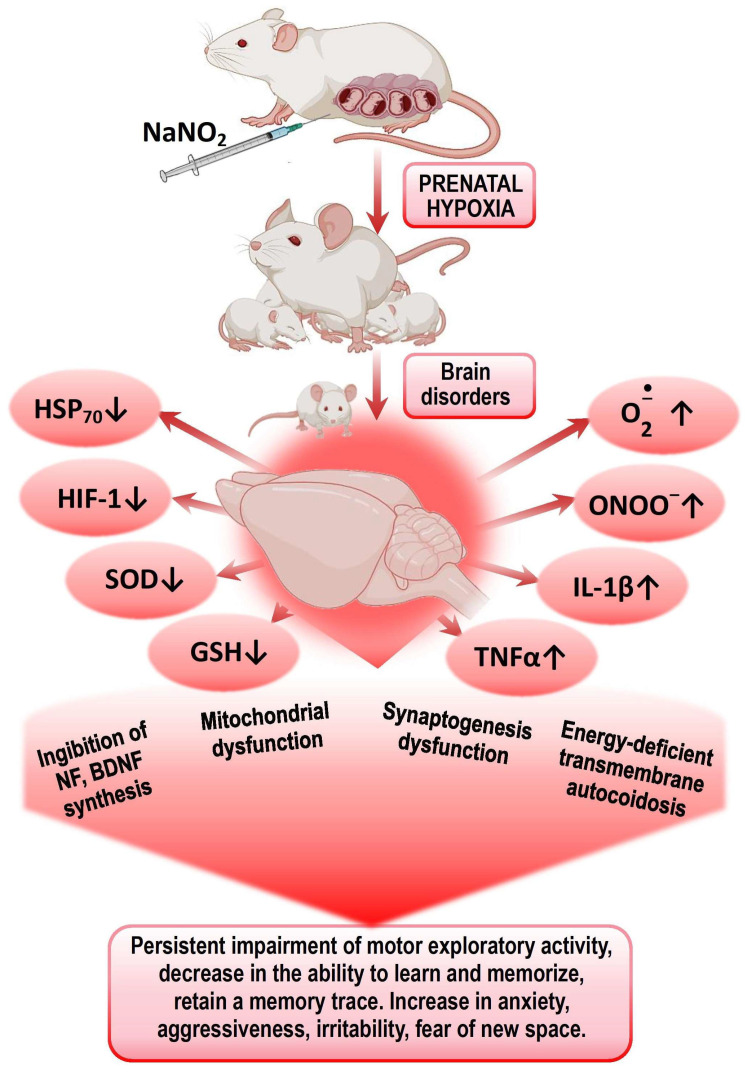
Mechanisms of formation of psychomotor and cognitive–mental disorders in offspring after PH.

**Figure 2 biomedicines-13-00982-f002:**
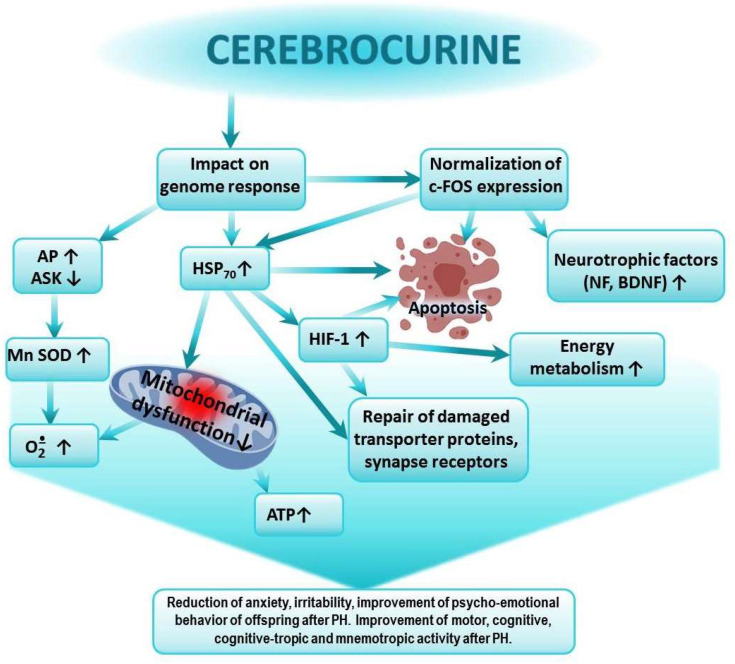
Possible mechanism of the positive effect of Cerebrocurin on cognitive-tropic functions of the CNS and on research, motor, and emotional activities of offspring after PH.

**Figure 3 biomedicines-13-00982-f003:**
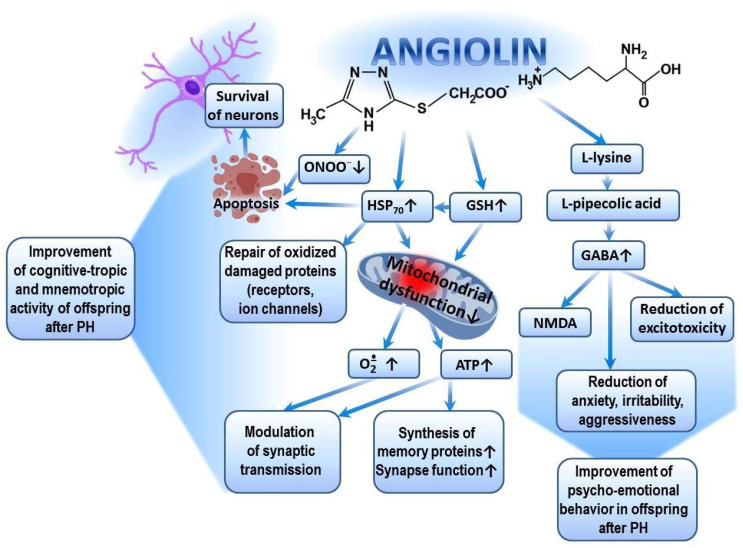
Possible mechanism of cognitive-tropic, mnemotropic and anti-anxiety effects of Angiolin after PH.

**Table 1 biomedicines-13-00982-t001:** The effect of the studied pharmaceutical agents on the exploratory activity of 1-month-old rats in the open field test following prenatal hypoxia (mean ± SEM, n = 10). Differences were considered statistically significant at *p* < 0.05 (Kruskal–Wallis test with Dunn’s post hoc correction).

Experimental Group	Overall Activity (cm^2^/s)	Time Spent Entering the Center (Sec.)	High Activity (%)	Inactivity (%)	Freezing (Count)	Free Distance (cm)
Intact	24,380.9 ± 1242.4	12.9 ± 2.4	17.8 ± 1.44	30.4 ± 6.7	282 ± 32	60.2 ± 4.3
Control PH	18,167.4 ± 1043.5 ^1^	28.3 ± 3.2 ^1^	4.3 ± 0.57 ^1^	82.3 ± 6.3 ^1^	561 ± 22 ^1^	29.7 ± 2.1 ^1^
PH + Cerebrocurin	45,762.2 ± 2286.5 ^1,2^	15.4 ± 1.5 ^1,2^	18.2 ± 1.55 ^2^	52.2 ± 4.2 ^1,2^	307 ± 17 ^1,2^	78.3 ± 3.5 ^1,2^
PH + Angiolin	39,863.2 ± 1022.5 ^1,2^	12.2 ± 1.7 ^1,2^	14.7 ± 1.00 ^1,2^	37.3 ± 3.4 ^2^	272 ± 12 ^1,2^	74.7 ± 1.7 ^1,2^
PH + Piracetam	27,952.1 ± 1103.0 ^1,2^	21.7 ± 1.2 ^1,^	8.7 ± 0.67 ^1,2^	54.3 ± 3.3 ^1,2^	412 ± 32 ^1,2^	50.2 ± 2.6 ^1,2^
PH + L-arginine	18,722.2 ± 978.2 ^1^	27.7 ± 4.1 ^1^	4.7 ± 0.72 ^1^	87.3 ± 8.2 ^1^	532 ± 34 ^1^	32.4 ± 3.2 ^1^
PH + Tamoxifen	17,764.4 ± 1103.3 ^1^	31.7 ± 3.5 ^1^	3.8 ± 0.25 ^1^	121.3 ± 15.7 ^1,2^	475 ± 35 ^1,2^	22.4 ± 2.5 ^1^
PH + Glutaredoxin	29,221.2 ± 992.2 ^1,2^	23.5 ± 2.7 ^1^	5.8 ± 0.35 ^1^	78.3 ± 4.7 ^1^	549 ± 34 ^1^	53.4 ± 4.2 ^1,2^
PH + Thiotriazoline	21,822.2 ± 1121.1 ^1,2^	21.4 ± 2.4 ^1^	7.8 ± 0.45 ^1^	73.4 ± 5.3 ^1^	501 ± 27 ^1^	46.7 ± 2.3 ^1,2^
PH + Mexidol	19,612.2 ± 1231.2 ^1^	27.3 ± 2.5 ^1^	4.3 ± 0.57 ^1^	87.8 ± 7.2 ^1^	447 ± 22 ^1,2^	44.2 ± 2.7 ^1,2^
PH + HSF1	30,573.4 ± 911.2 ^1,2^	18.0 ± 1.4 ^1,2^	8.5 ± 0.74 ^1,2^	77.3 ± 6.2 ^1^	423 ± 26 ^1,2^	62.2 ± 2.2 ^1,2^
PH + Mildronate	24,876.2 ± 914.2 ^2^	26.5 ± 2.2 ^1^	4.8 ± 0.65 ^1^	67.3 ± 5.3 ^1^	511 ± 67 ^1^	65.1 ± 1.4 ^1,2^

^1^—changes are significant in relation to animals of the intact group (*p* < 0.05); ^2^—changes are significant in relation to the animals of control group.

**Table 2 biomedicines-13-00982-t002:** The effect of the studied pharmaceutical agents on the exploratory activity of 1-month-old rats in the open field test following prenatal hypoxia (cont.) (mean ± SEM, n = 10). Differences were considered statistically significant at *p* < 0.05 (Kruskal–Wallis test with Dunn’s post hoc correction).

Experimental Group	Distance Along the Wall (cm)	Standing Next to the Wall (Count)	Short Grooming (Count)	Defecation (Count)	Immobility (Count)
Intact	4012.4 ± 277.5	4 ± 1	4 ± 1	3	282 ± 25
Control PH	5857.2 ± 205.2 ^1^	8 ± 1	1 ± 1 ^1^	1 ^1^	523 ± 17 ^1^
PH + Cerebrocurin	4521.2 ± 182.2 ^1,2^	4 ± 1 ^2^	4 ± 1 ^1,2^	4 ^2^	211 ± 12 ^2^
PH + Angiolin	4211.3 ± 234.2 ^1,2^	4 ± 1 ^2^	4 ± 1 ^1,2^	4 ^2^	280 ± 25 ^2^
PH + Piracetam	5922.3 ± 177.3 ^1^	5 ± 1 ^2^	2 ^1^	2	242 ± 18 ^2^
PH + L-arginine	5768.5 ± 187.3 ^1^	8 ± 2 ^1^	1 ^1^	1 ^1^	515 ± 22 ^1^
PH + Tamoxifen	6045.1 ± 312.7 ^1^	6 ^1,2^	1 ^1^	1 ^1^	577 ± 23 ^1^
PH + Glutaredoxin	5743.5 ± 197.4 ^1^	8 ^1^	1 ^1^	1 ^1^	473 ± 21 ^1^
PH + Thiotriazoline	5634.3 ± 223.8 ^1^	6 ^1,2^	1 ^1^	1 ^1^	435 ± 21 ^1^
PH + Mexidol	5433.2 ± 211.3 ^1^	6 ^1,2^	3 ^1^	1 ^1^	507 ± 32 ^1^
PH + HSF1	5245.3 ± 231.3 ^1^	6 ^1,2^	3 ^1^	1 ^1^	312 ± 22 ^2^
PH + Mildronate	5723.5 ± 311.6 ^1^	8 ^1^	1 ^1^	1 ^1^	251 ± 12 ^2^

^1^—changes are significant in relation to animals of the intact group (*p* < 0.05); ^2^—changes are significant in relation to the animals of control group.

**Table 3 biomedicines-13-00982-t003:** The effect of the studied pharmaceutical agents on the exploratory activity of 1-month-old rats in the open field test following prenatal hypoxia (cont.) (mean ± SEM, n = 10). Differences were considered statistically significant at *p* < 0.05 (Kruskal–Wallis test with Dunn’s post hoc correction).

Experimental Group	Number of Reference Memory Errors (Count)	Number of Working Memory Errors (Count)
Intact	2	5
Control PH	4 ± 1 ^1^	16 ± 1 ^1^
PH + Cerebrocurin	2 ^2^	7 ^1,2^
PH + Angiolin	1 ^1,2^	7 ± 1 ^1,2^
PH + Piracetam	3 ^1^	14 ^1,2^
PH + L-arginine	5 ^1^	16 ± 2 ^1^
PH + Tamoxifen	4 ^1^	14 ± 1 ^1^
PH + Glutaredoxin	4 ^1^	12 ^1,2^
PH + Thiotriazoline	3 ^1^	12 ^1,2^
PH + Mexidol	3 ^1^	15 ^1^
PH + HSF1	3 ^1^	10 ^1,2^
PH + Mildronate	4 ^1^	16 ^1^

^1^—changes are significant in relation to animals of the intact group (*p* < 0.05); ^2^—changes are significant in relation to the animals of control group.

## Data Availability

The original contributions presented in this study are included in the article.

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
