# Peer review of "HSP70 Modulators for the Correction of Cognitive, Mnemonic, and Behavioral Disorders After Prenatal Hypoxia"

_biomedicines, 2025, doi:10.3390/biomedicines13040982_

Round 1
Reviewer 1 Report
Comments and Suggestions for Authors
- Introduction is lengthy and can be condensed
- HSP70 modulators have been used earlier and as such the concept is not new. But authors have used more drugs in this study
- Hypoxia is reduction of oxygen content in the blood/ tissue. Asphyxia is a better term which indicates the damage caused by hypoxia.
- Placental insufficiency among human fetus causes growth restriction in addition to organ damage. Did the authors observe restriction in the growth among the offsprings?.
- How many offisprings were used after delivery and whether the animals had comparable gestation and weight ?. How many offsprings were available from Pregnant mices?. Did the use all available offsprings for the study?
- How many doses of HSP70 modulators were given and at how many days after birth?
- Did the investigators look for any toxicity on the animals from the drugs?
- Did the authors look for any anatomical changes in the brain among controls and experimental animals?
Author Response
We thank the Reviewer for the evaluation and feedback.
Q1. Introduction is lengthy and can be condensed.
A1. Thank you for the suggestion.
Q2. HSP70 modulators have been used earlier and as such the concept is not new. But authors have used more drugs in this study
A2. This article presents material from several years of research conducted by our team on developing approaches to neuroprotection following intrauterine hypoxia. We have proposed the concept of endogenous neuroprotection and the role of HSP70 in the mechanisms underlying this system, as reflected in our previous publications. We have selected HSP70 modulators as potential neuroprotective agents. This information is outlined in the introductory part of our article.
Q3. Hypoxia is reduction of oxygen content in the blood/ tissue. Asphyxia is a better term which indicates the damage caused by hypoxia.
A3. Regarding the terms "hypoxia" and "asphyxia," hypoxia refers to an oxygen deficiency in the body, while asphyxia is a form of suffocation of varying severity that occurs as a result of hypoxia. In mild cases, asphyxia can lead to hypoxemia (low oxygen levels in the blood) and acidosis (a condition where there is an excess of acid in the blood). Asphyxia is a pathological condition where the breathing process is disrupted, resulting in oxygen deficiency in the newborn. It may occur immediately after birth or within the first few days. Intrauterine hypoxia (the most common cause of neonatal asphyxia) can lead to asphyxia in newborns. Many researchers and textbook authors use the term "hypoxia" when referring to this pathology. Therefore, we insist on the use of the term "prenatal hypoxia".
Q4. Placental insufficiency among human fetus causes growth restriction in addition to organ damage. Did the authors observe restriction in the growth among the offsprings?
A4. Yes, in the group of untreated animals after perinatal hypoxia (control), we observed higher early mortality, as well as growth and development delays, which we have already reported in our earlier works.
Q5. How many offisprings were used after delivery and whether the animals had comparable gestation and weight ?. How many offsprings were available from Pregnant mices?. Did the use all available offsprings for the study?
A5. Each series of experiments consisted of 10 animals, all of which were matched in terms of gestational age. We used all surviving animals in the experiment, with the highest mortality observed in the control group. This is in line with our previous studies.
Q6. How many doses of HSP70 modulators were given and at how many days after birth?
A6. The doses were calculated in our previous experiments, as well as those conducted by other researchers, with corresponding references provided. The drugs were administered immediately after birth, in accordance with clinical neonatal resuscitation practices.
Q7. Did the investigators look for any toxicity on the animals from the drugs?
A7. We investigated the acute and chronic toxicity of Angiolin as a new molecule ((S)-2,6-diaminohexanoic acid 3-methyl-1,2,4-triazolyl-5-thioacetate). Based on the acute toxicity study results (in mice, rats, and rabbits), Angiolin was classified as a Class V substance (practically non-toxic) with no cumulative properties, as per the accumulation index. The drug does not exhibit skin-irritating effects on intact rat skin, local irritative effects on the intact mucous membrane of guinea pig eyes, allergic reactions in guinea pigs, or ulcerogenic effects in rats.
Additionally, after 180 days of intragastric administration of Angiolin at doses of 100, 500, and 1000 mg/kg, it was found that the drug does not cause structural or functional changes, does not lead to dystrophic or hemodynamic disturbances, and does not induce destructive reactions in the studied tissues of the animals.
Q8. Did the authors look for any anatomical changes in the brain among controls and experimental animals?
A8. Yes, the anatomical changes in the brain of control and experimental animals were investigated by us. In particular, we have conducted and are continuing morphometric studies of the CA1 hippocampus in the offspring after perinatal hypoxia and in the context of pharmacotherapy with HSP70 modulators. We have calculated neuron density, their area, RNA concentration, and the density of apoptotically altered cells. The density of neurons was reduced, the RNA level in neurons was also lower after PH. Moreover, apoptosis level increased following hypoxia. The drugs affected the hippocampal neurons after hypoxia, especially prominent effects were observed using angiolin and cerebrocurin (unpublished, in preparation).
Reviewer 2 Report
Comments and Suggestions for Authors
Dear authors,
Please highlight the novelty of your work in the abstract and introduction sections.
Kindly add recent references to the discussion.
I suggest you if key results of your work if you can in the form of a bar graph. It will enhance the reader's interest.
I suggest you highlight the current problems in this field and future outcomes of your work.
Is there any role of nanotechnology/or nanoformulation in future? kindly give your views how we can incorporat it.
Author Response
We thank the reviewer for evaluation and comments.
Q1. Please highlight the novelty of your work in the abstract and introduction sections.
Q2. Kindly add recent references to the discussion.
A1,2. We have made additions and corrections to the manuscript, accordingly.
Q3. I suggest you if key results of your work if you can in the form of a bar graph. It will enhance the reader's interest.
A3. The authors prefer to use the tabular format, as it provides absolute figures.
Q4. I suggest you highlight the current problems in this field and future outcomes of your work.
A4. We are planning further in-depth studies on the nasal gel with Angiolin as a promising neuroprotective agent. We hope to implement it in clinical neurology, neonatology, and resuscitation practice in the future.
Q5. Is there any role of nanotechnology/or nanoformulation in future? kindly give your views how we can incorporat it.
A5. We are currently working on a project focused on the development of drugs based on nanovanadium. The active ingredient of this drug is the trace element vanadium, which, as a result of innovative technologies, has acquired new properties that positively influence molecular (related to ROS) mechanisms of cellular signaling and metabolism. We plan to use it for the rehabilitation of male reproductive health.
Reviewer 3 Report
Comments and Suggestions for Authors
The article addresses the important issue of the impact of prenatal hypoxia (PH) on nervous system development, leading to cognitive and emotional disorders. The authors investigate the neuroprotective potential of various HSP70 modulators, analyzing their effects on motor activity, behavior, and memory in the offspring of rats subjected to PH. However, the article requires some revisions to improve clarity, methodology description, and data interpretation:
- Lines 102-103 – Was the approval granted by the Bioethics Committee or the Ethics Committee?
- How many experimental animals (offspring) were in the study and control groups?
- When were the tested substances administered, for how long, and according to what administration schedule?
- What were the housing conditions of the animals during the experiment (temperature, humidity, etc.)?
- Line 152 – What does the notation "i/p." mean?
- Line 164 – Is "overall motor activity" expressed in cm²/s or rather in cm/s?
- Please check the descriptions of groups 1, 2, 8, and 10 – if the administered substances are given in µL/g, this represents volume rather than the actual dose of the substance.
- Do the results present the mean for the group, and does the ± value refer to SD or SEM? Please clarify this in the results description and in the data analysis section.
- When describing results please add references to specific tables.
- I suggest presenting the most important parameters in graphical form – this would significantly improve interpretation.
- In the tables, please include the measurement units for the studied parameters in parentheses within the column headers.
- Tables should include a legend specifying: the number of animals per group, how the results are expressed (mean ± SD or SEM), the statistical tests used, the significance level applied, and an explanation of any abbreviations used in the table.
- Line 310 – Are you sure that Glutaredoxin reduced immobility? The table does not show a significant result.
Author Response
We thank the reviewer for the evaluation and feedback.
Q1. Lines 102-103– Was the approval granted by the Bioethics Committee or the Ethics Committee?
A1. Yes, this is now also stated in the Materials and Methods section of the study.
Q2. How many experimental animals (offspring) were in the study and control groups? T
A2. Ten animals per group. This has been now added to the manuscript in the tables.
Q3. When were the tested substances administered, for how long, and according to what administration schedule?
A3. This is now stated in the Materials and Methods section of the study. Intraperitoneally, using a micro syringe, once a day, from day 1 to day 30, immediately after birth.
Q4. What were the housing conditions of the animals during the experiment (temperature, humidity, etc.)?
A4. We have now included this section in the manuscript.
Q5. Line 152– What does the notation "i/p." mean?
A5. It is an abbreviation for 'intraperitoneal administration'. We have now removed it from the text, as it was redundant.
Q6. Line 164– Is "overall motor activity" expressed in cm²/s or rather in cm/s?
A6. It is in cm²/s, the area covered per unit of time.
Q7. Please check the descriptions of groups 1, 2, 8, and 10 – if the administered substances are given in µL/g, this represents volume rather than the actual dose of the substance.
A7. We have now corrected it and added explanations to the manuscript text.
Q8. Do the results present the mean for the group, and does the ± value refer to SD or SEM? Please clarify this in the results description and in the data analysis section.
A8. We have presented the data as 'mean ± standard error' (Mean ± SEM).
Q9. When describing results please add references to specific tables.
A9. Added.
Q10. I suggest presenting the most important parameters in graphical form – this would significantly improve interpretation.
A10. The authors prefer to retain the tabular data, as it provides absolute values.
Q11. In the tables, please include the measurement units for the studied parameters in parentheses within the column headers.
A11. Added.
Q12. Tables should include a legend specifying: the number of animals per group, how the results are expressed (mean ± SD or SEM), the statistical tests used, the significance level applied, and an explanation of any abbreviations used in the table.
A12. Added.
Q13. Line 310– Are you sure that Glutaredoxin reduced immobility? The table does not show a significant result.
A13. Thank you for the comment, it was a typo, it is now corrected.
Round 2
Reviewer 1 Report
Comments and Suggestions for Authors
- Authors have answered the questions but not included all informations in the revised manuscript.
- Perinatal asphyxia is an important cause of neonatal mortality and later disability. Perinatal period includes antenatal and early postnatal. Authors have induced hypoxia during the antenatal period. How long the induced hypoxia persists (only prenatal or continue till the birth)?
- Damage caused by Perinatal asphyxia does not continue after the inital phase (antenatal and during the first few days after birth). It is only the manifestations of damage seen later . It does not cause later neurodegenration and hence the first sentence in the abstract requires modification.
- Authors need to clarify whether all the offsrings used were similar in weight and gestation since insult and response to therapy will be different.
- Conclusions can be restricted to the main outcome (point numbers; 1,2 and 5 may be sufficient)
Author Response
We thank the Reviewer for the feedback and evaluation overall.
Q1. Authors have answered the questions but not included all informations in the revised manuscript.
A1. We have now reviewed the manuscript again and added additional information (highlighted in yellow).
Q2. Perinatal asphyxia is an important cause of neonatal mortality and later disability. Perinatal period includes antenatal and early postnatal. Authors have induced hypoxia during the antenatal period. How long the induced hypoxia persists (only prenatal or continue till the birth)?
A2. Modeling prenatal hypoxia by administering sodium nitrite to pregnant females leads to hemic hypoxia due to the formation of methemoglobin, which combines with tissue hypoxia due to the uncoupling of oxidation and phosphorylation processes. Disruption of oxygen transport function in the blood of pregnant female rats results in impaired uteroplacental circulation and oxygen starvation of the fetus or embryo. Administration of sodium nitrite at a dose of 50 mg/kg from the 15th to the 21st day of pregnancy leads to hypoxic encephalopathy of varying severity in adult individuals when exposed to a hypoxic factor during the prenatal period of development, according to the criteria proposed by N.F. Ivanitskaya. This is described in our work (Belenichev, I.; Popazova, O.; Yadlovskyi, O.; Bukhtiyarova, N.; Ryzhenko, V.; Pavlov, S.; Oksenych, V.; Kamyshnyi, O. Possibility of Using NO Modulators for Pharmacocorrection of Endothelial Dysfunction After Prenatal Hypoxia. Pharmaceuticals 2025, 18, 106. https://doi.org/10.3390/ph18010106). This information is now included in the manuscript.
Q3. Damage caused by Perinatal asphyxia does not continue after the inital phase (antenatal and during the first few days after birth). It is only the manifestations of damage seen later. It does not cause later neurodegenration and hence the first sentence in the abstract requires modification.
A3. There are works available in the public domain that describe the role of prenatal hypoxia in the development of brain functions in the postnatal period and the subsequent increase in the risk of neurodegenerative disorders at later ages. Prenatal hypoxia has a significant impact on the expression and processing of various genes involved in the normal functioning of the brain, as well as their epigenetic regulation. Prenatal hypoxia leads to changes in the expression patterns of mRNA and proteins, such as acetylcholinesterase and amyloid precursor protein, amyloid-β peptide. Disruption of their expression and metabolism, caused by prenatal hypoxia, may also lead to the development of neurodegeneration at later ages, in addition to early cognitive dysfunctions. We have already mention this risk of neurodegenerative pathology after prenatal hypoxia in the abstract, and thus we kindly request that the abstract remains as it is.
Q4. Authors need to clarify whether all the offsrings used were similar in weight and gestation since insult and response to therapy will be different.
A4. Yes. We have now included this information in the Materials and Methods section.
Q5. Conclusions can be restricted to the main outcome (point numbers; 1, 2 and 5 may be sufficient)
A5. The answer: We have now made the changes in the text accordingly.